# Beehive Smart Detector Device for the Detection of Critical Conditions That Utilize Edge Device Computations and Deep Learning Inferences

**DOI:** 10.3390/s24165444

**Published:** 2024-08-22

**Authors:** Sotirios Kontogiannis

**Affiliations:** Laboratory Team of Distributed MicroComputer Systems, Department of Mathematics, University of Ioannina, University Campus, 45110 Ioannina, Greece; skontog@uoi.gr; Tel.: +30-26510-08208

**Keywords:** precision apiculture, decision support systems, IoT, embedded systems, agriculture 4.0, classification algorithms, convolution neural networks, neural networks, sound analysis, bee colony collapse disorder, deep learning, edge computing

## Abstract

This paper presents a new edge detection process implemented in an embedded IoT device called Bee Smart Detection node to detect catastrophic apiary events. Such events include swarming, queen loss, and the detection of Colony Collapse Disorder (CCD) conditions. Two deep learning sub-processes are used for this purpose. The first uses a fuzzy multi-layered neural network of variable depths called fuzzy-stranded-NN to detect CCD conditions based on temperature and humidity measurements inside the beehive. The second utilizes a deep learning CNN model to detect swarming and queen loss cases based on sound recordings. The proposed processes have been implemented into autonomous Bee Smart Detection IoT devices that transmit their measurements and the detection results to the cloud over Wi-Fi. The BeeSD devices have been tested for easy-to-use functionality, autonomous operation, deep learning model inference accuracy, and inference execution speeds. The author presents the experimental results of the fuzzy-stranded-NN model for detecting critical conditions and deep learning CNN models for detecting swarming and queen loss. From the presented experimental results, the stranded-NN achieved accuracy results up to 95%, while the ResNet-50 model presented accuracy results up to 99% for detecting swarming or queen loss events. The ResNet-18 model is also the fastest inference speed replacement of the ResNet-50 model, achieving up to 93% accuracy results. Finally, cross-comparison of the deep learning models with machine learning ones shows that deep learning models can provide at least 3–5% better accuracy results.

## 1. Introduction

The applications of deep learning (DL) and machine learning (ML) algorithms are quickly expanding toward the agricultural sector, offering new capabilities and assisting in overcoming several barriers to existing solutions. Towards this direction, the expansion of the IoT industry and the cloud digitization process extensively affects livestock management and, specifically, apiaries, offering solutions to problems that could not be solved without continuous repetitive beekeeping observations.

The appliance of automated control procedures and targeted interventions against critical beehive conditions is the beekeeper’s main concern and is carried out through periodic control checks of the entire apiary. In the context of smart automation of the Beekeeping Industry 4.0, many innovative solutions of cloud telemetry, decision support, data collection, and condition control are being designed and implemented at the European level [1,2]. These Decision Support Systems observe, albeit timorously, through intelligent machine learning or deep learning detection and prediction algorithms to facilitate the beekeeper and limit unnecessary interventions [3].

Deep learning (DL) models embedded in IoT devices, offering precise predictions and detections, are considered the next step toward Agriculture 4.0. Those devices can transmit measurements or compute threshold-based correlation functions and provide model inferences using a time series of sensory features as input. Dense input measurements close to real-time are needed to achieve better than best-effort detections and forecasts. Over the last decade, several deep learning (DL) models have been developed to assist beekeepers in detecting catastrophic beehive conditions due to illnesses and stress and guide them in their practices. Using IoT sensory inputs and deep learning inferences, precision beekeeping transforms into a conditions management tool of localized appliances and tools of artificial intelligence [4]. Enhancement with digital twins will further augment beekeeping practices, supported by deep learning models risk classification, productivity, mite/parasite disease object detection, and predictive interventions [5].

Apiculture deep learning (DL) models target sustainable bee colony growth, resilience to critical conditions, precise time planning, and suggestive interventions. Complicated equations or machine learning approaches such as linear regression, k-NN, or Support Vector Machines [6] are slowly set aside due to the increased training data volumes and features. In contrast, deep learning models such as Convolutional Neural Networks (CNNs) use convolutions for attribute reduction combination and multi-layered or multi-stranded neural networks for decision making, instantiating either cloud or edge device inferences [7,8].

Dense measurement inputs are needed to achieve better than best-effort classifications and forecasts. Only then can the deep learning-trained models provide more accurate results than their machine learning counterparts. The simplified approach regarding model data inputs, outputs, and trainable hyperparameters should also be mentioned, as it is required so that the big data input measurements, data transformation, and training processes are easy to apply and enforce. Such an easy-to-apply methodology will eventually lead to repetitive auto-encoding and labeling tasks that can provide a set of different pre-trained models, generating inferences of various types of detection under different environmental conditions and prerequisites.

In order to provide accurate and early apiculture threat detection or abnormalities that can be of assistance to the beekeepers to perform targeted interventions, this paper presents a new IoT detection device called Bee Smart Detection (BeeSD) device, a modification of the device previously presented in [9], that additionally includes device embedded deep learning models. BeeSD is a low-cost beehive cloud transponder of temperature–humidity sensory measurements, raw sound recordings and edge computing detections. The previous BeeSD implementation presented in [9] suffered from the use of the inaccurate DHT-11 temperature–humidity sensor that failed to precisely capture the humidity conditions inside the beehive and the use of only a single-point DS18B20 temperature sensor from Maxim Integrated Inc., San Jose, CA, USA. This single-point temperature probing could not provide multi-point temperature measurements among frames to detect colony growth or identify appropriate conditions for disease incubation. Therefore, the presented Bee Smart Device, apart from its embedded deep learning inference capabilities, includes the low-cost, more accurate DHT-22 temperature–humidity sensor from Aosong Electronics Co., Ltd., Guangzhou, China, to measure conditions at the beehive lid and up to six DS18B20 temperature sensor probes to acquire sensory measurements among different frames. Therefore, this study aims to develop and evaluate the BeeSD real-time edge device for monitoring critical beehive conditions using advanced deep-learning techniques.

The remainder of this paper is organized as follows. Section 2 presents a systematic review of the available literature on machine learning–deep learning emerging technologies for detecting critical beehive conditions. Section 3 presents the new Bee Smart Detection prototype implementation, proposed edge computing detection process, application-level protocols, and interfaces. Section 4 presents the experimental scenario and evaluation metrics used. Section 5 presents the experimental results. Finally, Section 6 concludes this paper.

## 2. Related Work

The most devastating events for the beekeeping colony (hive) today do not concern exogenous threats but events occurring within the hive, such as (1) the occurrence of extreme conditions and a lack of food in the settlement area (low to zero bee colony growth) [10,11], (2) the birth of a new queen (queen cells), mainly in spring and autumn, which leads to the phenomenon of swarming, i.e., the abandonment of the colony by the new queen(s) with part of the population [12], (3) the incubation of diseases (mainly the varroa mite and nosemiasis (Nozema ceranae) [13,14,15], which are the main destructive diseases for apis mellifera in Europe, (4) human environmental interventions (pollution and use of pesticides) [16], and (5) the mass destructive major colony loss, which affects a significant portion of beehive colonies (Colony Collapse Disorder), mainly due to combined factors of diseases, lack of nutrition, and extreme conditions inside the hive [17,18].

More specifically, CCD incidents are suspected as a multi-factor event of exposure to fungal diseases (N. ceranae), mites (varroa destructor), lack of colony nutrition, and nonlethal pesticide exposure [17,18,19,20]. As a result of CCD events, significant population losses emerge, leading to a minimum resistance to diseases, extreme environmental conditions, and/or maintaining vital but stressful conditions inside the beehive (minimal thermoregulation capabilities, extreme levels of humidity); thus, monitoring the beehive’s response to stress can detect such CCD events early [21].

These catastrophic CCD cases may endanger the current beekeeping production and the entire beekeeping hive (beekeeping capital), requiring increased efforts for continuous surveillance, interventions, and preventive measures by beekeepers, even the most experienced ones, to avoid the abovementioned risks. Therefore, one of the most important tasks in beekeeping involves constantly observing conditions inside and outside the hive. Security systems successfully assist the beekeepers by offering automated monitoring and control outside the beehive (inspection cameras, electric fences, GPS IoT nodes). Therefore, beekeepers’ efforts are concentrated on periodic inspection to ameliorate catastrophic events. The beekeeping tasks currently concentrate on observing the beehive’s conditions. Such observations include the opening of the beehive, removal of the frames, observing the queen and the deposition of eggs and larvae growth, observing the workers laying honey, removing possible queen cells, visually locating the queen, and performing visual inspection for diseases (varroa, clipped wings of workers, nosema fungus signs, frame rot).

Beekeepers’ problems in their inspections include diseases, stress, and signs of Colony Collapse Disorder (CCD events), mainly due to poor nutrition, polluted environments, and extreme environmental conditions. In contrast to the other branches of animal husbandry, where livestock owners can easily find a qualified veterinarian to help them deal with their problems, beekeeping is more complicated. To sustain the beehive colony and keep it solid and healthy, the beekeeper must have a profound knowledge of bee diseases, stress detection, and treatment methods. The most significant diseases that require immediate intervention are the nosema ceranae and varroa mite diseases. At the same time, stress conditions include swarming and queen loss due to aging or environmental conditions (low temperatures below 5 °C and above 40 °C) and lack of nutritious resources, specifically pollen. In this section, the author focuses on new technological IoT systems [22] and smart Decision Support Systems that use IoT devices for precision beekeeping [23] and utilize machine learning (ML) and deep learning (DL) algorithms, which in turn can be used for detecting and reporting critical condition cases [4,24]. A detailed description of the most catastrophic apiary events follows.

The varroa mite disease is one of the most common causes of bee mortality. The varroa mite is one of the destructive diseases if found at population levels above 3–5% and, if not treated, it will cause colony degradation [13]. Varroa mites may incubate in beehives, spread across them, weaken bee colonies’ population and productivity, and finally lead to collapse within a year [25]. Numerous types of research have been carried out; algorithms [4,7] using IoT devices similar to [26] have been implemented, which take as input data images and aim at the early detection of varroa via image processing or object detection. Additionally, another varroa mite detection approach was presented that does not directly use images as input data but provides the ability of an on-demand measurement to detect varroa mite infection in the hive using object detectors [27]. Similarly, [28] promotes the use of an IoT device incorporating classifiers of both machine learning models (Decision Trees, Extreme Gradient Boosting (XGBoost)) and a deep learning custom NN model, which use measurements of temperature, humidity, and CO_2_ or TVOC (Total Volatile Organic Compounds) with a defined level of contamination by the parasite for CO_2_ concentration of 1000–2000 ppm and TVOC of 600–1500 ppb, which vary in proportion to the temperature and humidity values inside the beehive.

Nosema ceranae is another catastrophic disease caused by a fungal parasite that affects the bee’s digestive system and causes significant losses, specifically in autumn. It is a tissue parasite responsible for the short lifespan of honey bees and less egg laying by the queen [29]. Nosema is a fungus; therefore, it is hard to detect inside the beehive. Only lab-based detection methods, such as the Polymerase Chain Reaction study of the Nosema spp., have been developed, as well as primers within the framework of primer specificity and sensitivity [30], throughout microscopy, chromatography [31], and electrochemical detection methods [32]. Nevertheless, an image detection method using a portable CMOS with camera zoom capabilities acting as an amplifying microscope and UV light emitting diodes as an excitation light source has been proposed as a quantifying portable instrument for the detection of Nosema spores [33]. There are still no field machine or deep learning methods for detecting this disease. Nevertheless, for the case of nosemiasis detection, a camera system that uses image processing algorithms and neural networks in controlled laboratory environments has been proposed by [34].

CCD beehive condition threats can also be detected by monitoring beehive sounds. Different audio signatures and patterns are emitted by bees under stress or illness excitation, and they are considered to have significant detection confidence, such as Varroa mite reproduction, predator attacks, the absence of the queen, bee swarming, food availability, and exposure to harmful environmental conditions [35]. Among these threats, we highlight the absence of the bee queen since a healthy queen in full posture is critical for the hive. Therefore, detection of the presence of the queen bee is critical because it acts as a warning signal to the beekeeper, indicating an irreversible situation and the need for major intervention actions. Nolasco et al. proposed an IoT solution to detect queen bees through audio processing and machine learning [36]. Machine learning and sound analysis were also used for the same reason by Cejrowski et al. [37]. Furthermore, using Neural Networks and SVM (Support Vector Machines) can also offer queen detection from sound recordings [38,39]. Furthermore, another approach for queen bee presence detection includes using deep learning models [40].

Honey bee starvation due to a lack of nutrition is another CCD threat for bees. Starvation may be caused by unfavorable weather, disease, long-distance transportation, or depleting food reserves. Over-harvesting of honey (lacking supplemental feeding) is the foremost cause of scarcity, as bees need more honey and pollen stored. However, weather, pollution, and the use of pesticides can also cause problems. Frameworks have been developed, and systems have been used for beehive health monitoring. Thus, they can detect harmful events such as beehive starvation [41,42,43]. Therefore, the bee population needs to be periodically inspected. Towards this direction, Zhang et al. [44] experimented with custom MLP models and Hierarchical Generative Prediction Networks that take as input both sensory (temperature, humidity, pressure) and sound data to show the potential of using custom deep learning audio models for predicting strengths of beehives based on multi-modal data composed of audio spectrograms and environment sensing data. Also, Fernando et al. presented a multi-temperature probing IoT device of 40 temperature sensors and a placement method to represent the inner status of the beehive using heat maps [45]. Other ways of detecting the lack of nutrition or population reduction are the use of external weight scales [46] and regression [47] or deep learning LSTM [48] models.

Swarming is also another serious event. Swarming is a stress-initiated event where new queens are born, and a large proportion of the population abandons the hive following the new queen. It occurs during the spring when the abundance of food from flowering allows the leaving swarm to survive. The preparation for the swarming starts up to 20 days before, and until then, the colony has to ensure that its queen (who will lead the swarm away from the hive) can lose weight so she can fly [49]. Therefore, the bees responsible for feeding (nurse bees) stop feeding the queen, so she stops spawning (laying eggs) due to the change in diet. Then, the open larvae, which need nourishment, are reduced daily. This condition alters the smooth functioning of the hive concerning the collection of pollen and nectar, as the remaining bees limit their work more and more, day by day, until new egg production is also halted. This forces the bees to flap their wings (buzzing) at a frequency of about 250 Hz [50]. This event increases in intensity over time until the day of swarming.

Machine learning and deep learning models have been used to detect swarming. Zakepins et al., in their research on the detection of swarming, implemented an IoT device that uses temperature sensors and studied the variation in temperature values in the internal environment of the hive and observed during the warm-up period for the final take-off; the temperature above the upper body of the hive increases by 1.5–3.4 °C above the normal temperature of the brood [24]. Furthermore, Kiromitis et al. also implemented an appropriate IoT device constructed for collecting sound data, and the classifiers used for this purpose were the SVM, k-NN, and half U-Net [6] in a primer version of the BeeSD device [9]. Another study proposed a new beekeeping condition monitoring system to detect bee swarming. The proposed system included two versions of the end node devices and a new algorithm that utilizes CNN deep learning networks [26].

Sound recordings are mainly used to detect queen loss events. Several methods have been examined in the literature [4]. Cejrowski et al. [37] mentioned that the bee worker sounds in the air are at 250Hz and estimated the band in which a microphone would be sensitive to be 20–20 kHz. In their IoT implementation, they added the microphone, temperature, and humidity sensors to a bee frame. In their implementation, they used Linear Predictive Coding to acquire coefficients that describe the signal and SVM to detect queen loss. Nolasco et al. [36,51] presented a queen loss detection technique using minute audio recordings split into 2 s slices of 22.05 KHz samples and filtered frequencies between 0 and 6000 Hz, generating Mel spectra of 120 bands and MFCCs of 20 coefficients and also using the Hilbert Huang Transform (HHT). Then, two approaches have been compared to their performance to distinguish queen loss: the SVM algorithm and a custom CNN model of four layers. From their experimentation, the SVM over MFCCs has been proven to be more accurate, achieving an AUC score [52] of 0.91–0.94, more than their proposed CNN model, which did not manage to generalize well in the presence of unseen hives. Terenzi et al. [53] presented several feature extraction methods apart from HHT, such as Continuous/Discrete Wavelet Transform (CWT-DWT) from bee sound, and their importance in detecting orphan colonies (queen-less). From this study experimentation, both STFT and CWT models achieved the best mean recall values for the queen/no-queen classes. In-depth, the STFT model achieved mean recall values of 96%, and the CWT model achieved mean recall values of 92.5%.

Regarding sound recordings and swarming, Kiromitis et al. [6] performed a similar experiment for the detection of swarming, using 10 s audio recordings of 22.05 KHz sampling rate, between 0 and 10 KHz, generating Mel spectrogram images and comparing the Mel response classifier for swarming events, using a deep learning Half-UNet network and a k-NN and SVM ML methods for the early (5–10 days before the event) and late detection (5 days before the event) of swarming. From their experimental results, the SVM achieved accuracies from 90% for early detection to 97% for late detection, and k-NN achieved the best score for late detection at 98% and for early detection at 85%. At the same time, the CNN half-UNet achieved an 89% score for late detection (poor performance close to the best-achieving SVM) and an accuracy for late detection of up to 95%. Libal et al. [54] also experimented with the detection of swarming by using a Short Time Fourier Transformation (STFT), Mel Frequency Cepstral Coefficients (MFCCs), also calculating their first and second derivatives delta Δ and delta delta $\Delta^{2}$ and using Lasso regularization to detect imposing MFCC values from a feature set. Then, an auto-encoder neural network is used to minimize the mean square error (MSE) of one class, thus maximizing the error of the other, a specified adaptive MSE calculated value above a training threshold value. This is to classify worker bee and drone classes and detect higher activity of male drone bees close to the beehive entrances. The best accuracy achieved by this binary classifier was 95.3%.

Summarizing the literature results regarding the use of sound for detecting queen loss and swarming, it is evident that a machine learning model such as SVM can achieve overall mean accurate detections close to 90%. However, DL algorithms performed better than ML algorithms in cases of early detection events (swarming, queen loss).

Collecting multi-parameter data on beehive conditions is essential for accurately detecting CCD events. To this end, Hong et al. [55] presented a new beehive design that incorporates a holistic conditions monitoring IoT system that utilizes an in-hive weight sensor, sound sensor, and photoelectrical bee counter at the beehive entrance as a holistic conditions monitoring system. The beekeepers use weight scales to identify the availability or scarcity of food and detect the initiation of swarming events (loss of at least 1 kg of weight). However, their work does not present significant experimental results nor incorporate machine or deep learning event detection methods, apart from telemetry monitoring and threshold-based alerts. Moreover, acquiring weight measurements in the beehive rather than with an external weight scale can be a powerful tool for detecting the lack of food in the surrounding environment and initiation of swarming events. To this extent, Bellos et al. [22] also presented a holistic IoT monitoring device implemented on Langstroth beehives, using temperature, humidity, in-hive load sensors, gyroscopic sensors, magnetic reeds, and an RGB camera using image preprocessing at the device level and low-energy LoRaWAN protocol for data transmissions.

Furthermore, focusing on the alleviation of stressful condition events, Kontogiannis [56] also presented an IoT approach for the automated confrontation of extremely low- or high-temperature and humidity instances inside the beehive using Peltier coolers and thermopads as well automated ventilation openings controlled by servo motors. His proposition has been implemented into typical Langstroth beehives. Nevertheless, his implementation requires significant amounts of energy to provide thermoregulation and, therefore, can be implemented only on static apiary installations.

Finally, apart from the mentioned systems, there are several IoT system implementations and commercial systems [56], monitoring conditions using sensors such as temperature, humidity, air, and external weight scales. However, most of those IoT system implementations serve as supportive tools for visualization, providing telemetry monitoring and threshold-based alerts via services and interfaces. Therefore, they offer limited use of machine learning or deep learning methods for detecting events. The author would like to comment especially on the BeeLive system proposed by Hamza et al. [57], which utilizes the open-source ThingsBoard Application Server and the MQTT telemetry protocol also used by the author for the implementation of its BeeSD IoT device cloud services and interfaces.

Monitoring the acoustic frequency response inside a beehive under normal conditions, environmental stress (high temperatures and/or increased humidity levels), due to poor nutrition, and before a swarming event, the following frequencies have been identified, as illustrated in Figure 1. As mentioned by [58], amplitude frequencies between 400 and 500 Hz indicate swarming activity, as mentioned by Ferrari et al. at 300–500 Hz. Moreover, according to [9], two frequency response areas are also detected: swarming-A area, where increased response values signify the incubation of new queen(s) cell(s), and swarming-B area, with increased response values for a swarming event, that is, a swarming event about to happen (birth of a new queen bee).

Transforming beehive sound frequency response to images and then using deep learning CNN classification models can achieve significant detection results. By performing selective feature reduction and reducing the number of trainable parameters to a set of frequency responses produced from the beehive, data inputs of significant sizes are processed via convolutions flattened down to dense neural networks. Classifier models, such as VGG-16 or VGG-19 (Visual Geometry Group) Convolutional Neural Networks (CNN) [65], multilayered ResNets [66] of convolution and skip-connection identity blocks, and Inception multi-filter convolutions and filter concatenation block networks [67,68], are the commonly used classifiers for object detection tasks. These models have already scored the top 5 accuracy scores in the Imagenet dataset [69]. This paper focuses on microprocessor edge implementations of the VGG-16 [70], VGG-19 [71], ResNet-18, and ResNet-50 [72], and downsampling only half-UNet models [73]. The Inception v3 model implementation has also been examined due to the ResNet model similarities [74] and multi-filter implementation diversion concerning ResNet models. Furthermore, the variable depth and widen factor WideResNet model is also examined as an alternative variant for ResNet [75]. The author presents its new BeeSD device implementation classifier for detecting queen loss and swarming events from sound frequency responses transformed to images and utilizing well-known deep learning CNN models for this task at the IoT device level in Section 3.3.

Multipoint temperature measurements inside the beehive are used to detect conditions that may cause stressful events (CCD). This is better achieved with the use of multi-temperature probes placed in different frames inside the beehive (usually one with larvae and another with bees), as well as the use of a separate temperature and humidity sensor placed in the beehive lid (inner beehive surface), measuring temperatures at the beehive inner top as well as beehive humidity. Extreme humidity measurements can be set above 75% relative humidity (RH). Extreme high-temperature measurements are above 37 °C for probe temperature measurements and above 42 °C for lid temperature, while extreme low-temperature measurements are below 10 °C for probe temperature measurements and below 5 °C for lid temperatures. The values mentioned above are taken initially from the literature [12] and re-calibrated from the author’s observations for three years in the beekeeping station (by monitoring winter measurements and beehive mortality and by monitoring summer conditions and beehive growth stills). Winter conditions are considered more dangerous than the summer since, at such temperatures, the bees might not have enough protein fat to sustain the colony (winter bees), and/or the colony might freeze to death.

To achieve CCD condition detection, classes of extreme conditions have been constructed by the author: (1) class of abiotic cold (low temperature, high humidity, LTHH), (2) class of low temperature and low humidity (LTLH), (3) class of normal conditions, (4) class of high temperature–low humidity (HTLH), and (5) abiotic class of high temperature and high humidity (HTHH). From the author’s classification, only two classes are considered critical and characterized as abiotic. The author proposes a new process for detecting CCD abiotic conditions using multi-point temperature and humidity measurements via the fuzzy-stranded-NN model [76,77]. The author presents the new BeeSD device embedded stranded-NN classifier model for detecting favorable CCD conditions in Section 3.3.

## 3. Materials and Methods

In this section, the authors present his BeeSD IoT device and system proposition components and capabilities. The proposed BeeSD system can detect bee starvation, CCD conditions, swarming events, and queen-loss events based on two deep learning (DL) processes that provide detection inferences at the device level (BeeSD edge computing IoT nodes). The BeeSD system’s high-level architecture description follows.

### 3.1. BeeSD High-Level Architecture

A new Bee Smart Detector (BeeSD) system is proposed for monitoring and detecting major critical beehive events. The proposed system uses IoT devices called smart BeeSD nodes, an edge computing alternative of [9], which uses multiple temperature probes and more accurate sensors, capable of measuring temperature, humidity, and raw audio data with the use of a lavalier microphone inside the beehive. The BeeSD nodes also utilize Wi-Fi gateways to transmit sensory measurements, can generate FFT spectrograms, FFT-responses of specific frequency ranges of sound spectrogram data called beegrams, and detection inferences of queen loss, swarming, and critical conditions. Data outputs and inferences are transmitted to the BeeSD cloud logging service, provided by the ThingsBoard Application Server (AS) [78], utilizing two distinct communication channels: (a) HTTP requests to an open-source ThingsBoard Application Server (AS) [78] for short-length data transmissions less than 500 KB, and (b) MQTT publishes for big data transmissions of raw sound data, no more than 200 MB for cloud model training purposes. Figure 2 illustrates the proposed Bee Smart Detection system, components, and capabilities.

The BeeSD system includes the following components:**BeeSD end node device:** This is the end node IoT device placed inside the beehive (see Figure 2(1), BeeSD node). The BeeSD IoT device can operate in two modes: training and detection. In training mode, the device periodically transmits 20 to 45-minute raw sound measurements and hourly temperature and humidity measurements. Training data transmissions are performed via (a) MQTT publish over the MQTT broker (see Figure 2(4)) and then are stored to the BeeSD file server (see Figure 2(5)) for later processing and model training. Usually, during training mode, the sound recordings are performed at night to avoid both daily noises and stress conditions caused to the beehive by human interventions or animals during the day. Moreover, to increase the system security and data integrity, each BeeSD IoT device is equipped with a VPN key in order to connect to the cloud services and transmit training data, authenticated via the VPN service (see Figure 2(3)). The BeeSD end node IoT device design and prototype are illustrated in Figure 3.In detection mode, the BeeSD device transmits sensory measurements (see Figure 4a), Mel spectrogram images (see Figure 4b), major FFT parameters and attribute values (called beegram values; see Figure 4c), which correspond to the significant frequency bands presented in Figure 1, mean aggregated beegram values (see Figure 4d), which correspond to beehive growth levels, queen tone, thermal stress, and swarming indications, detection inferences for queen loss and swarming acquired by the CNN image detection models, and CCD conditions detection calculated by the fuzzy-stranded-NN model (see Figure 2(f), detection model inference). These values are transmitted over HTTP telemetry posts to the ThingsBoard Application Server (AS) [78] (see Figure 2(6)). Suppose a model update parameter is set at the AS. In that case, the BeeSD IoT device may also download an updated version of a specific pre-trained model used for either detecting swarming and queen loss or critical conditions (see Figure 2(e), HTTP GET model request).**BeeSD gateway:**  This is the BeeSD system gateway (see Figure 2(2)), placed in the center of the beekeeping area, and it includes a Wi-Fi-to-3G/LTE access point gateway device with its corresponding 12 V–100 Ah battery, 20 W photovoltaic panel, and voltage regulator to provide autonomous uninterruptible operation. This gateway connects all BeeSD IoT nodes to the BeeSD system’s cloud services.**Data logging and visualization services:** The BeeSD system also performs measurement logging and visualization in detection mode and measurement logging in training mode. The ThingsBoard AS performs detection mode logging and visualization tasks (see Figure 4). Measurements are stored in its Cassandra database, while IoT device management and dashboard operations are stored in its PostgreSQL database (see Figure 2(5)). Therefore, ThingsBoard’s core functionality is for storing data measurements sent by the BeeSD IoT nodes via HTTP POSTs, offering pre-trained model updates to the IoT nodes, and dashboard visualizations via HTTP GET requests to the beekeepers. The training mode data logging is performed using different MQTT channel topics (MQTT publish) over an MQTT broker (see Figure 2(4)). These trainable data are then stored in trainable data files (images, sensory measurements) in the BeeSD file server (see Figure 2(5)).**AI component:** The BeeSD system AI component (see Figure 2(7)) is a dockerized container including all necessary libraries and training tools for performing deep learning model training. The container has access to the BeeSD File server via HTTP requests (see Figure 2(b)); to obtain all necessary data needed for model training, it can perform data filtering, fuzzy annotation, CNN model training, and fuzzy-stranded-NN model training. The training process is performed on demand and manually by the system’s supervisor (see Figure 2(c)). The supervisor is also responsible for uploading pre-trained models as per BeeSD device server parameters to the ThingsBoard AS using HTTP POSTS (see Figure 2(d)).**Mobile phone Application:** The BeeSD system mobile phone application is an end-user application capable of visualizing ThingsBoard HTTP dashboards. It is a web progressive Android application that the beekeeper uses to connect to the ThingsBoard AS and monitor his device (per device) dashboards (see Figure 2(g) and dashboard illustrations in Figure 4). That is, each device is identified as a ‘hive-id’–‘node-id’ tuple. Then, for each device, a specific dashboard is manually generated (using ThingsBoard plugins and IoT device telemetry measurements) (see Figure 4) and assigned to each beekeeper. This way, the beekeeper can monitor the measurements of his/her devices, beegram attribute values, beegram correlation functions values, spectrogram, listen to audio recordings, and monitor the DL notifications and alerts for swarming events, queen loss, and critical abiotic conditions.

The following subsection describes in detail the BeeSD IoT device, parts, and smart functionality.

### 3.2. Bee Smart Detection IoT Device

The BeeSD IoT node device used by the BeeSD system is a Raspberry Pi Broadcom BCM2710A1 64bit quad-core ARM cortex at 1GHz, equipped with 512MB DDR2 RAM and 32GB of flash storage, where the Linux Debian OS system resides with 4GB of swap space and Python version 3.9 installed (see Figure 3(a7)). In its GPIO interface (see Figure 3(a9)) are connected at least two (up to six) one-wire DS18B20 temperature probes (see Figure 3(a4)), exiting the device via two electrical wire seals (see Figure 3(a2)), to perform point temperature measurements in between the beehive frames. A DHT22 temperature–humidity sensor (see Figure 3(a10)) is placed a few inches out of the IP56 plastic case (see Figure 3(a11)). The device has two plastic openings to screw in the inner surface of the beehive lid (see Figure 3(a8)). The DHT22 temperature measures the temperature levels close to the lid. In contrast, the DS18B20 sensors measure the temperature between frames of the same or different brood box floors. The BeeSD device includes a lavalier microphone connected to the RPi USB port (see Figure 3(a1)). The microphone is placed outside the device and tied up at the center of the beehive lid. The BeeSD device also includes a Wi-Fi transponder (see Figure 3(a8)) and an I2C on/off circuit (see Figure 3(a8)) connected to the 12V battery pack (see Figure 3(a3)), as described in the previous BeeSD version, implemented in [9]. The battery pack is placed on the beehive lid, under the 12V PV panel and the voltage regulator (see Figure 3b).

This BeeSD IoT node is an updated version of the device presented in [9], equipped with at least two DS18B20 sensors instead of one to provide multiple points of probing inside the beehive and a DHT22 temperature humidity sensor of better temperature resolution of ±0.5% than DHT11’s ±2% and better humidity accuracy of ±2% than DHT11’s ±5%, as well as an expected humidity measurement range of 0–100% instead of the 5–95% RH of the DHT11 sensor. The new, improved Bee Smart Detection device is connected via a 12-to-5 V 1A step-down converter attached to its power cable directly from a 12V PV charge regulator. The charge regulator is attached at the back of a 10–20 W panel and a 12 V–8 Ah Lithium battery (see Figure 3b).

The most significant feature of this new IoT end node device is its ability to perform edge computations and deep learning inferences than its predecessor [6,9]. It can generate and compute from raw sound data, FFT spectrograms, Mel spectrograms (see Figure 4b), and generation of beegrams (mentioned in Section 3.3; see Figure 4c,d) with the use of the SciPy and scikit-learn libraries [79,80]. It can perform deep learning inferences using CNN models using the PyTorch framework [81]. It can perform deep learning inferences using the fuzzy-stranded-NN model [76,77] implemented in the TensorFlow [82] framework. It also has the necessary pre-trained CNN models to provide queen loss and swarming detection from sound recordings and CCD condition detection from sensory measurements via the fuzzy-stranded-NN model. Moreover, the newly improved BeeSD edge detection device is more accurate and more autonomous than its predecessor since it can detect apiary destructive events offline by monitoring apiary conditions and executing device-level inferences.

### 3.3. BeeSD DL Detection Process

For the process of detecting swarming events and queen loss, the BeeSD IoT device includes the PyTorch framework [81] and CNN models (examined in Section 4) to extract such information from recorded sound data in detection mode. Such detection inferences are usually based on 1-minute audio recordings performed at night on a per-hour period. Five distinct classes reside in the pre-trained CNN models:Class 0:Queen loss class. It corresponds to the queen loss class, trained using audio recordings of forced queen removal events performed in selected beehives in the author’s beekeeping station over the years 2023 and 2024. This annotated dataset contains 14GB of audio recordings (approximately 100 h of queen loss audio recordings performed in four beehives corresponding to 5882 JPEG spectrogram images).Class 1:Weak bee class. This class corresponds to the annotated class recordings of weak bees (under five frames inside the beehive) that are in the phase of growing or under stress since they have newly born queens. This dataset has been recorded in three young beehives during spring and two during late autumn of 2022–2023. It contains 23.3 GB of audio recordings (approximately 167 recorded hours, corresponding to 10,020 min JPEG spectrogram images).Class 2:Normal bee class. It corresponds to the annotated class recordings of normal beehives (strong beehives of at least ten frames per beehive). This dataset was acquired during the spring and summer seasons of 2023–2024. It contains 17.3 GB of audio recordings (approximately 126 recorded hours, corresponding to 7560 min JPEG spectrogram images).Class 3:This class is the early swarming detection class. It was trained using audio recordings of beehives approximately ten days before the first swarming event. This dataset was acquired during forced swarming events out of four 10-frame beehives in the spring and summer seasons of 2022–2023. It contains 12 GB of audio recordings (approximately 86 recorder hours, corresponding to 5160 min JPEG spectrogram images).Class 4:This is the late swarming detection class. It was trained using audio recordings of beehives five days before the swarming event. This dataset has been acquired during forced swarming events out of 10-frame beehives in the spring and summer seasons of 2022–2023, out of seven beehives. It contains 16 GB of audio recordings, approximately 115 recorded hours, corresponding to 6900 min JPEG spectrogram images).

The inference process of the CNN DL classifier models initiates with the recording of at least 30 s (30 s–90 s) of raw WAV file using the BeeSD IoT device lavalier microphone (8-bit PCM unsigned values with a sampling rate of 22,050 Hz and max frequency range of 11 KHz). Then, the raw WAV data are processed by a low-pass filter of 2.5 KHz bandwidth (0–2.5 KHz) (see Figure 5(1b)). The recorded WAV file is then split into smaller chunks using a 10-second window. The first 10 s split of the recording is discarded due to low-frequency microphone startup noise harmonics. In these recorded WAV files, the device performs a Sparse FFT transformation. This calculation creates matrices of 513 × 862 elements for every 10 s window. The size of 513 corresponds to the segment’s 0–11 KHz frequencies, while the value of 862 corresponds to the 10 s spectrogram interval. Then, the spectrogram image is generated using Mel values instead of frequencies, logarithmic scale amplitude values, and a 20 s window. For 30 s of the recordings, the last 20 s are used to generate the spectrogram image. As a rule, the 0.2X − 0.2X + 20 s is used for X-minute recordings. Only the 20 s of the raw data recording is converted to a spectrogram image and is sent to the ThingsBoard AS via HTTP POST (see Figure 5(1e)).

The generated five matrices (1 min recording) by the SFFT process of 513 × 862 elements are each further narrowed down to 224 × 224 matrices, maintaining the first 224 values on the x- and y-axes of the original ones (which correspond to the frequencies of up to 5 KHz and 0–5.3 s time intervals). These five 224 × 224 matrices’ magnitude values are first averaged (mean elementwise amplitude response). Then, the calculated mean response matrix A (224 × 224) of x elements is normalized to 0–1: $\frac{x-\bar{A}}{max(A)-min(A)}$, and then its elements are multiplied by 255 to represent a gray-scale image and converted to a single-channel image (see Figure 5(1g)), passed to the DL CNN model for classification.

The beegram attributes comprise a 16-element vector b[0],b[1],…,b[15]. Each vector element corresponds to a specific frequency range mean magnitude (frequency batch) of the first 31 frequency response elements of the 224 × 224 matrix A, generated by the SFFT spectrogram of the raw audio recordings. For example, b[0] corresponds to $\bar{A[4,:]}$, that is, mean magnitude value response in the frequency range of 107–128 Hz, b[1] corresponds to the mean magnitude value response in the frequency range of 129–149 Hz, and so forth, up to b[15], which corresponds to the mean magnitude response of the frequency range of 602–687 Hz. All these mean responses correspond to a specific bee frequency response area, as denoted in Figure 1 (mentioned in Section 2) and illustrated in Figure 4c. These beegram parameters are calculated (see Figure 5(1d)). Then, from these attributable parameters, specific measures of interest are calculated (see Figure 5(1f)) and sent along with the parameters by the IoT device to the ThingsBoard AS via HTTP POST. These specific measures correlation functions are presented in Table 1 and include indicator values of hunger (Bh), growth (Bg), queen tone (Qt), thermal stress (Th), and swarming events (Sw). The measure value limits (presented in Table 1) have been calculated and calibrated using the author’s recorded dataset.

For detecting critical CCD conditions inside the beehive that may lead to CCD events, the BeeSD IoT device includes the TensorFlow framework [82] and the fuzzy-stranded-NN model [76,77], with 96-value batch input size, which corresponds to daily temperature and humidity measurements of a BeeSD IoT device that includes two temperature probes and its external (beehive lid) DHT22 temperature and humidity sensor (four hourly measurements for 24 h, equal to 96 observation points of input batch size 96). The trained fuzzy-stranded-NN model includes trained strands of 72, 96, 120, 144, 168, and 192 batch sizes that correspond to devices of 1–6 temperature probes accordingly and the calculation of daily inferences. The batch measurements are entered in a specific order, starting with the temperature probe values, followed by the DHT22 sensor temperature and humidity values ($T_{p1}T_{p2}TH$). Five distinct classes exist in the trained strands:Class 0:It is the abiotic cold condition class. This class includes lid temperatures below 0 °C and mean probe temperatures below 10 °C. This critical condition case requires immediate intervention (beehive relocation to a warmer place, use of passive insulation, use of active thermal actuating systems [56]) to avoid CCD winter bee stress and freezing events of imminent colony death. For this class, humidity measurements have no significant influence.Class 1:It is the disease condition incubation class. It includes humidity measurements above 80% RH for mean temperature values below 20 °C ($10<\bar{T}\leq 20$) and humidity measurements above 90% for mean temperatures above 20 °C ($20<\bar{T}\leq 28$). In this class, appropriate interventions should occur (check for varroa mite and nozema fungus concentrations in the frames, for wingless bees, and for internal water leak concentrations in the brood box). These class conditions are also triggered by multiple days of continuous rainfall.Class 2:It is the normal condition class. This class includes probe temperature differences, specifically lid–probe temperature differences no more than 1–6 °C, where the maximum temperature probe value is between 30 and 37 °C. In this class, humidity values are between 30 and 80% of RH. Typical normal temperature probe values are around 34–36 °C and humidity levels are between 40 and 65% of RH values.Class 3:It is the abiotic hot condition class. It includes relative humidity values below 75% RH, lid temperature values above 42 °C, and temperature differences less than 3–4 °C between probes (mean temperature value) and lid (DHT22). This critical condition usually occurs after a swarming or heat stress event in a weak bee population. It requires beehive relocation, water installments near the beehive, specialized interventions of larvae frame addition taken from other beehives, or even queen removal (replacement). Especially if mean temperature probe values are above 37 °C, it is a critical indication of a catastrophic collapse (inability to thermoregulate the hive).Class 4:It is the outlier/erroneous measurement class. It includes all measurements not fitting into one of the classes above. Generally, this class includes erroneous measurements due to broken or misplaced probes, measurements of an abandoned (empty) beehive, or measurements that do not feel like beehive colony measurements.

The dataset for training the fuzzy-stranded-NN model used the previous single-probe version of the Beesd IoT device [9]. For class 0, winter measurements of three-week beehives were used, exposed to winter conditions that led to colony loss (CCD) in 2023 (a total number of 1120 ($T_{p},T,H$) tuple measurements). For class 1, three beehives exposed to summer conditions after multiple swarming events in 2022 were used, as well as two weak bees that collapsed during autumn of 2023 (a total number of 7727 ($T_{p},T,H$) tuple measurements). For class 2, five strong beehives (of at least ten frames) were used, taking measurements during spring, summer, autumn, and winter of 2022–2024 (a total number of 16,000 ($T_{p},T,H$) tuple measurements). For class 3, stressed temperature conditions of 3 beehives prior to the swarming events of 2022 were used to signify abiotic hot stress and two beehives during the summer seasons of 2023–2024 (a total number of 640 ($T_{p},T,H$) tuple measurements). Finally, the class 4 sample was randomly generated from 1000 ($T_{p},T,H$) erroneous measurements. Classes 0 and 1, and three datasets were randomly generated and then fuzzy annotated to the appropriate class until they reached 16,000 measurements as of class 2. Figure 6 shows the fuzzification process used for data generation for classes 1, 2, and 3, respectively.

Before the training process of the fuzzy-stranded-NN model, the collected dataset’s fuzzy annotation process was instantiated to normalize the dataset, remove outlier values, and provide logical annotations for generated temperature and humidity values. The four classes that have been used denoting the low-temperature abiotic and high-temperature abiotic conditions, using bounded sigmoid functions, can be expressed by the following Equation (Equation 1):(1)$$bs\left(x\right)=\frac{1}{1+e^{x\frac{4\cdot ln(3)}{x_{low}-x_{high}}}\cdot 9e^{x_{low}\frac{-4\cdot ln(3)}{x_{low}-x_{high}}}}$$
where $x_{low}$ and $x_{high}$ are the bounded temperature values where the sigmoid function is close to 0 and 1. For the cold abiotic class, an inverse bounded sigmoid function is used (1−bs(x)) with parameters $x_{low}=0$ and $x_{high}=10$. For the hot abiotic class, a sigmoid function is used with parameters $x_{low}=38$ and $x_{high}=42$. A triangular sigmoid function has been used for the disease and normal classes, expressed by Equation (Equation 2).
(2)$$ts\left(x\right)=\left\{\begin{array}{cc} bs(x) & ifx\leq x_{center},\left(x_{low},x_{high}=x_{center}\right)\\ 1-bs(x) & ifx>x_{center},\left(x_{low}=x_{center},x_{high}\right) \end{array}\right.$$
where $x_{center}$ is the center value of this function, where $ts(x_{center})≊1$ and $x_{low},x_{high}$ values are the bounded function values, where $ts\left(x_{low}\right)=ts\left(x_{high}\right)≊0$. For the disease class, the tuple $(x_{low},x_{center},x_{high}$ values were set to (10, 15, 28). For the normal class, the tuple $\left(x_{low},x_{center},x_{high}\right)$ values were set to (15, 32, 38). Figure 6a illustrates the temperature fuzzy sets and their corresponding confidence level values over x-axis temperatures in three distinct humidity case values of 40%, 60%, and 90% RH.

For the humidity fuzzy sets, the three sets correspond to dry, normal, and wet humidity values. For dry values, an inverse bounded sigmoid (1−bs(x); see Equation (Equation 1)) function is used with parameters $x_{low}=0$ and $x_{high}=30$. For the wet value class, a sigmoid function is used with parameters $x_{low}=70$ and $x_{high}85$. A Gaussian function has been used for the normal value class, expressed by the following Equation (Equation 3).
(3)$$g\left(x\right)=e^{-b{\left(x-x_{0}\right)}^{2}}$$
where $x_{0}$ is the peak of the graph equal to the value of 40, and *b* defines the steepness, and it is set to 0.004. Figure 6b illustrates the humidity class values over humidity (x-axis). Finally, Figure 6c illustrates the fuzzy annotated results over different values of temperature and humidity set to 40, 60, and 90%, respectively. The fuzzy rules used for acquiring such fuzzy class inference results are presented in Table 2. The “plus” notation corresponds to 25% more influence of that class, whereas the “minus” notation corresponds to 25% less influence of that class (in parentheses).

The inference process of the fuzzy-stranded-NN model is initiated by recording daily temperature–humidity measurements. These measurements are initially stored in the SD card of the BeeSD IoT device. Upon collection, the data are batch-partitioned and sent to the appropriate model strand based on the number of temperature probes used (see Figure 5 flowchart (2), steps (a), (b), and (c)). The inference class result is then sent to the ThingsBoard AS via HTTP POST. Section 4 presents the author’s experimentation and experimental results using the fuzzy-stranded-NN 96 and 144 model strands for detecting abiotic conditions and different CNN models for detecting swarming and queen loss events.

## 4. Experimental Scenario

The author’s experimentation and deep learning process evaluation include the two BeeSD classification processes: the first for detecting swarming and queen loss via sound recordings and the other for detecting possible occurrence of CCD events using temperature and humidity measurements and the fuzzy-stranded-NN deep learning model. Appropriate data have been collected for the training process by provoking swarming events and queen loss events and recording stressful beehive conditions, especially in the winter and autumn, as described in Section 3.3 (per data annotation class). Those data are based on observations collected from 2018 to 2024 in the beekeeping station maintained by the laboratory team of Distributed MicroComputer Systems, Department of Mathematics, University of Ioannina, Greece (MCSL team, https://kalipso.math.uoi.gr) in the Ligopsa area of Ioannina, Epirus, Greece (lat: 39.791035, long: 20.654526). This dataset contains sound recordings and temperature and humidity measurements taken from the BeeSD device prototypes (eight implemented prototypes). For sound measurements, the data were recorded during the night. These recording preprocessing steps included splitting the data into 1-minute raw audio recordings and conversion to spectrogram images annotated appropriately depending on the observed-caused phenomenon (swarming, late–early, queen loss, weak bees). For the CCD stressful condition experimentation, the fuzzy-stranded-NN algorithm has been evaluated using collected temperature and humidity measurements, as described in Section 3.3, preprocessed accordingly by annotating and fuzzy filtering the temperature–humidity data batches. Figure 7 illustrates the beekeeping station in the Ligopsa area where the BeeSD prototypes have been tested.

The swarming and queen, loss model experimentation, includes evaluating several state-of-the-art CNN image classification models over 224 × 224 single-channel spectrogram sound images. The reason for using CNN models is to achieve optimal feature extraction and classification results via convolution layers provided by FFT spectrogram images (prior to the CNN model NN layer setup), provided by previously tested, significant accuracy results of CNN models on image classification problems. The algorithms tested are VGG-16, VGG-19, ResNet-16, ResNet-50, WideResNet-40, Inception_v3, and Half-UNet [6], in terms of accuracy, convergence speed, and loss. These CNN models have been implemented in the BeeSD IoT device, and their inference speeds have also been calculated. This experimentation aims to find the fastest CNN model (algorithm) that scores with the best accuracy and least training convergence. The training and testing dataset used was the one mentioned in Section 3.3. The metrics described in Section Evaluation Metrics have been used to evaluate CNN and fuzzy-stranded-NN models. A new metric called convergence speed has been introduced to measure how fast a CNN algorithm minimizes its classification loss value.

The CNN classifiers’ training process begins with an 80–20% image dataset split into training and testing sets. One hundred training epochs and a batch size of over 64 images per iteration are used. The Adam optimizer is the mathematical function used to update the model weights [83], using the default optimizer parameters of learning rate lr=0.001, first-second moment estimate exponential decay rate $\beta_{1}=0.9$ and $\beta_{2}=0.999$ and $\epsilon =10^{-8}$. The CNN model hyperparameters used were, for Inception_v3, a label smoothing factor of 0.1 for the cross entropy loss function, for ResNet-18 a stride of (2, 2) and a dropout layer of 0.2% drop probability, for ResNet-50, a stride of (1, 1) and zero dropouts, while for the WideResNet model, a depth of 40 was used, with a widen factor of 4 and zero dropouts. ResNet-18 dropout layers have been added to examine the effects of the dropout layers on the model and cross-compare the results with ResNet-50 of zero dropouts (data over-fitting control check).

The CCD detection experimentation includes evaluating two fuzzy-stranded-NN models (batch sizes with strands of 96 and 144 values) for detecting CCD events in terms of accuracy and convergence speed. The author uses the fuzzy-stranded-NN model on sensory measurements since there is no need to perform feature extraction due to the temporal measurements’ sparsity and the small measurement fluctuations over time. The fuzzy-stranded-NN model of different depths provides long and short memory depths similar to LSTM models. BeeSD IoT devices use the first model with two temperature probes, while BeeSD IoT devices use the latter 144-value batch size strand with four temperature probes. These models have been implemented in the BeeSD IoT device as strands of a stranded-NN model, and their device inference speeds have also been calculated. The fuzzy-stranded-NN model’s training process uses the fuzzy annotated dataset described in Section 3.3.

During fuzzy-stranded-NN training, an 80–20% dataset split into training and testing sets is performed during training. A total of 100 training epochs over 64 batches per iteration are used. The Adam optimizer is the mathematical function used to update the model weights [83], using the default optimizer parameters of learning rate lr=0.001, first-second moment estimate exponential decay rate $\beta_{1}=0.9$ and $\beta_{2}=0.999$ and $\epsilon =10^{-8}$. The fuzzy-stranded-NN model hyperparameters are zero dropouts and an L1 regularization (Lasso regression penalty to the loss function) per dense layer of λ=0.005.

### Evaluation Metrics

The two classification models have been tested in the evaluation scenarios: (1) the swarming queen loss classifier, which utilizes CNN models, and (2) the CCD classifier, which tries to detect abnormal in-hive sensory conditions using the fuzzy-stranded-NN algorithm. The mean average categorical cross-entropy loss and classification accuracy metrics have been used to evaluate the CNN models. Only the classification accuracy metric has been used during training on the testing datasets to evaluate the CCD classifier. Furthermore, a new metric called model convergence speed metric was used to measure the accuracy of the metric rate over training epochs. The metrics used are thoroughly described in the paragraphs below. Classification accuracy and categorical losses are commonly used in machine learning and deep learning classification algorithms to identify their classification potential. In contrast, the proposed convergence speed metric detects the number of required training epochs for a model to reach a maximum optimum accuracy result.

The categorical cross-entropy loss function is commonly used and measures the difference between the predicted class probabilities and the actual class labels for each testing set input. Categorical cross-entropy loss is calculated using Equation (Equation 4).
(4)$$L_{CE}=-\frac{1}{N}\sum_{i}^{N}\sum_{j=1}^{c}y_{ij}\cdot log\left({\hat{y}}_{ij}\right)$$
where *C* is the number of classes, $y_{ij}$ is the categorical value of the ground truth vector for sample *i*, and $y_{ij}=\left[y_{i1},\ldots y_{ic}\right]$ are the predicted values of sample *i* containing for each class j=1..c the normalized detection output values that express the per-class detection probability.

The classification accuracy is calculated as the total number of detected samples over the total number of samples. If $C_{nf}=\left[C_{ij}\right]$ is the n × n confusion matrix of the actual *i* and predicted *j* values of an annotated value set S, then the accuracy value is given by Equation (Equation 5).
(5)$$Acc_{S}=\frac{\sum_{1\leq j<i\leq n}C_{ij}}{\sum_{i=1}^{n}\sum_{j=1}^{n}C_{ij}}$$
where the numerator fraction equals the diagonal confusion matrix elements *C* and the sum of all confusion matrix *C* values in the denominator. Both accuracy and loss values are calculated during the training phase for every epoch as a mean of a validation set that corresponds to 10% of the training set and during the testing evaluation phase over the testing set. A new metric, called model convergence speed metric, is introduced to measure each algorithm’s training potential, which measures the accuracy rate over training epochs and is calculated using Equation (Equation 6).
(6)$$Conv_{n}=\frac{\partial Acc_{n}}{\partial e_{n}}=\frac{\sum_{i=e_{k},\;n>k}^{e_{n-1}}Acc_{i+1}-Acc_{i}}{n-k+1}$$
where $Acc_{i}$ is the testing accuracy value at epoch $e_{n}$ and *n* is the total number of examined epochs. The convergence speed metric value expresses how fast the model has reached its maximum accuracy value over training epochs. If convergence metric value $Conv_{n}\gg 1$, then the model quickly increases its accuracy value over epochs. If it persists, this is probably due to overfitting. If $1\geq Conv_{n}>0$, then the model is considered at a good training curve, increasing its accuracy. The same applies if $0>Conv_{n}\geq -1$, reducing its accuracy. Moreover, if $Conv_{v}\to 0$, then the model reached its maximum detection capability in terms of accuracy (maximum training epoch). Finally, if Conf≪−1, the model is set as an underachieving one that cannot fit the training data well (high bias and probably significant variance).

## 5. Experimental Results and Discussion

The BeeSD system evaluation cross-examines the accuracy and loss of several CNN classification algorithms for detecting swarming and queen loss as a unified model. It also includes the evaluation of the fuzzy-stranded-NN model for the detection of stressful conditions inside the beehive that may cause or be the cause of a CCD event. Appropriate data have been collected for the swarming and queen loss model training, and scenarios have been implemented. These data are based on observations at the beekeeping station maintained by the Laboratory team in the Ligopsa area of Ioannina, Epirus, Greece (lat: 39.791035, long: 20.654526). The data collected were selected and normalized on case-by-case scenarios of forced swarming events and queen removal.

The evaluation of the CNN and fuzzy-stranded-NN models initiates with the model’s training in the BeeSD system cloud AI container (see Figure 2(7)). Typical model sizes and the number of trainable parameters are presented in Table 3. Table 3 also presents the mean cloud inference speeds achieved by the models. The training cloud VM (Virtual Machine) used is an AMD 64-bit 24-core Virtual Machine processor with 64GB of RAM (60GB available for the training process and 64 GB of swap size). The model’s training has been performed at the CPU level with a minimum model training time (ResNet-18) of 7 days and a maximum model training time (WideResNet-40) of 23 days.

Focusing on cloud inference speeds of the CNN models, ResNet-18 is the model that achieves the least cloud inference times for a 224 × 224 input spectrogram image, followed by the ResNet-50 model, which takes 66.6% more inference time. Then, the Inception_v3 model follows, taking 137% more time than the ResNet-18 (570 ms), followed by the Half-UNet model, which takes 2.5 times more time than the ResNet-18 model (840 ms). The VGG and WideResNet models are considered slow inference models. The VGG-16 model takes about 3.8 times more time than the ResNet-18 model, which is closely followed by the VGG-19 model (4.79-5 times more than ResNet-18). The WideResNet model is the slowest in inference time, taking at least 13 times more time than the ResNet-18 model.

Examining the model sizes, which is a significant factor for device level edge computations, ResNet-18 and Inception_v3 CNN models have the smallest sizes (below 200 MB), followed by the ResNet-50 with a 71% bigger model size from the smallest one (ResNet18, 106 MB). The Half-UNet model follows with a model size of less than 500 MB (3.2 times bigger than the ResNet-18 model size). These are considered memory-resident models for BeeSD IoT edge devices with 512 MB of available RAM. They can remain in memory without disturbing the monitoring device’s operations. VGG models require more than the available device memory (830 and 884 MB, respectively), so they can only be loaded and unloaded, utilizing a significant portion of the IoT device swap area. The inference process of these models at the device level commands that all device monitoring operations be halted during inference tasks. Finally, the WideResNet-40 model size, being close to 2 GB, forces the IoT device operating system to kill its process during inference since it utilizes more than four times the device memory size. That is, the WideResNet-40 model cannot be used for edge computing inference but only for cloud computing inference tasks.

The fuzzy-stranded-NN model inference times are significantly less than the ResNet-18 inference time. Strand-96 achieves 37.5% less inference time, while strand-144 achieves 20% less inference time. Additionally, the stranded-NN model size, calculated as the sum of all strand sizes, is close to 2–12 MB (for 4–8 strands, respectively), which is very small compared to the CNN model sizes. Therefore, the fuzzy-stranded-NN model is considered a fast-loading resident model for IoT devices with at least 32 MB memory sizes.

The swarming and queen loss CNN models have been trained using 224 × 224 single-channel image representations of 10 s mean magnitude SFFT recordings of frequency responses up to 5 KHz (pre-filtered to 2.5 KHz using a low-pass filter). The maximum number of training epochs used was 100, and the mean achieved accuracy was calculated at three epoch breakpoints: 20 epochs, 50 epochs, and 100 epochs. The convergence speed metric values were calculated using frequencies in these breakpoint intervals. Table 4 shows all evaluated CNN models’ accuracy and convergence values over the mentioned breakpoint epochs. The same applies to the fuzzy-stranded-NN models used to detect CCD conditions.

From the evaluation results of the CNN classification models, the ResNet-50 model achieved the best mean accuracy scores overall training epochs concerning all models, followed by the WideResNet-40 model (0.8% less achieved accuracy) and the Half-Unet (3% less accurate). The Resnet-18 model achieved better mean accuracy results over all epochs than the VGG-16 (10.8% better accuracy) and marginally better mean accuracy results (0.46% better) than the VGG-19 model. The VGG-19 model’s mean accuracy results over epochs are considered to be 6.5% less than the mean score achieved by the ResNet-50 model. Finally, the Inception_v3 model has presented poor performance results. Due to the significant convergence speed values of the model and its complicated model structure compared to the other CNN models, more training epochs may be needed for it to perform similarly to the rest of the models.

The ResNet-18 model’s best mean accuracy results in overall epochs occurring at 50 epochs (95.71%), where its convergence speed metric value is minimal. Nevertheless, it is outperformed by the ResNet-50 at 50 and 100 epochs of at least 3% less mean accuracy. Half-UNet also outperforms ResNet-18 at 50 and 100 epochs by at least 0.6%. However, due to the less than 1% accuracy differences between the ResNet-18 and Half-UNet models, both can be considered models with similar accuracy results. Mean accuracies of the VGG-19 and VGG-16 models at 100 epochs are less than those achieved by the ResNet-18 model by at least 3.5% and 13.5%, respectively. Moreover, the VGG-19 convergence speed metric value is close to zero, meaning it will not improve its performance over more training epochs. At the same time, VGG-16 has a convergence value of 0.12 at 100 epochs, meaning there is still a window of improvement over training. In general, the VGG models failed to adequately detect swarming and queen loss events compared to the ResNet and the Half-UNet models. Their performance results are close to the ones achieved by other machine learning models, such as SVM and k-NN.

The WideResNet-40 model manages to outperform the ResNet-18 model both at 50 and 100 training epochs by at least 1.8% and 2.8%, respectively, in terms of mean accuracy over epochs, with a convergence speed close to zero. Nevertheless, it failed to perform better than the ResNet-50 model, presenting similar mean accuracy results to the ResNet-50 model at 100 epochs. Compared to the WideResNet-40 model size (close to 2 GB), this model is a bad candidate for edge computing inferences since it requires available memory sizes of at least 1 GB. Figure 8 presents the analytically achieved model’s accuracy values over trained epochs, while Figure 9 presents the models’ classification loss metric values over trained epochs.

From the accuracy results in Figure 8, it is obvious that the VGG-19 model, as well as the Inception_v3 models, are slow-learning models, with VGG-16 approaching the accuracy results of the rest of the models (VGG-19, Half-UNet, WideResNet-40, ResNet-50), but not minimizing its losses enough, while Inception_v3 fails to achieve more than 82% accuracy at 100 epochs and presents significant loss. More training epochs are probably required for this due to the complexity of operations between blocked layers. On the other hand, ResNet-18 with introduced dropout layers presents accuracy fluctuations and increased classification losses as epochs increase. The use of dropout layers in the ResNet-18 model was introduced as a strong over-fitting countermeasure concerning other regularization techniques. Nevertheless, it has been proven inefficient regarding accuracy results since it was expected to achieve smoother accuracy and loss curves over epochs, similar to the ResNet-50 model. The author notes that smoother results can be achieved if dropout layers are not used. The VGG-19 model classification loss also fluctuates at low training epochs, but it declines above 80 epochs, closely following the Half-UNet model classification losses. Finally, ResNet-50 and WideResNet-40, followed by the Half-UNet model, present the best accuracy over epoch curves, the maximum accuracy results over 100 trained epochs, and the minimum classification losses.

From the evaluation of the fuzzy-stranded-NN models (see Table 3), the 144 model strand that utilizes BeeSD IoT devices of four temperature probes achieved 1.6% better results in terms of accuracy over the 96 model strand that is used by the BeeSD IoT device that utilizes two probes. Nevertheless, the compensation of 1.6% improved accuracy results in contrast to the placement issues of four probes, placement efforts needed, as well as possible probe malfunctions of a multiple-probe device, making the device that utilizes only two probes a more efficient monitoring and CCD detection solution.

The author compared his swarming sound detection process with existing machine learning approaches mentioned in the literature [6], such as k-NN and SVM. From these comparison results, the best-achieving ResNet-50 deep learning model outperformed the machine learning counterparts by 2–12% for detecting late and early swarming events, respectively. More specifically, as mentioned in [9], the SVM model achieved 90% accuracy for early swarming detection and 97% accuracy for late swarming detection. In contrast, the k-NN algorithm achieved 85% accuracy for early swarming detection and 98% for late detection. Thus, the mean accuracy results achieved by the k-NN model are 91.5%, while for the SVM model, they are 93.5%. This means that all deep learning models except Inception_v3, VGG-16, VGG-19, and ResNet-18 (which performed similarly to the SVM model) outperformed their machine learning counterparts for swarming detection by at least 3% in terms of accuracy. Additionally, VGG-16 and VGG-19 presented overall similar results (less than 1% in terms of mean accuracy over epochs) and, in some cases, significantly better results, especially for the early detection cases of swarming. For early detection cases, all deep learning models (except Inception_v3) performed significantly better than the best case SVM detections by 1–7% in terms of mean accuracy results.

The author compared his queen loss detection process with the STFT and CWT deep learning models mentioned in the literature [53]. From the evaluation results, the best-achieving ResNet-50 model achieved 3% better accuracy than the STFT model (compared to the overall mean recall results) and 7% better accuracy than the CWT model. Additionally, comparing the accuracy results achieved by the authors in [28] with the ones achieved by the fuzzy-stranded-NN model for the detection of CCD conditions that are favorable for the growth of the varroa mite (disease-favorable environment), the fuzzy-stranded-NN-144 model achieved 95.2% accuracy results, 1.2% more than those achieved by the authors’ XGBoost model (mean accuracy of 94%). The author did not compare the [28] custom NN model with fuzzy-stranded-NN implementation since the authors need to provide adequate information on the model.

In summary, the author has to mention that his proposed CNN classification model, which combines both swarming and queen loss detection cases, is a new and innovative model approach. Furthermore, his best classification model, representative of ResNet-50, outperformed both deep learning and machine learning swarming classification models [6] and deep learning queen loss detection models [53]. Moreover, the fuzzy-stranded-NN model outperformed existing machine learning models for detecting CCD conditions. The combined use of these two models in the BeeSD IoT device promotes the implementation of the BeeSD as a holistic detection system with edge detection capabilities. These capabilities can be exploited by offering offline detection essential for migratory apiaries (migration locations of limited or nonexistent network coverage).

Regarding CNN and fuzzy-stranded-NN model inference speeds, as mentioned in Section 3.2, the BeeSD edge computing device used is an RPi-zero 1GHz 64-bit Arm Cortex-A53 quad-core with 512 MB DDR2 RAM, 32 GB storage size, and a 4096 MB storage cache (swap size) for big-size model load purposes. The OS used is the 64-bit Rasbian OS and Python version 3.9. Table 5 shows raw sound data transformation times to MEL spectrograms, beegram calculation, and 224 × 224 JPEG input image preparation time (indicated as data transform time) and CNN model load and inference times. Table 5 also shows the load and inference times of fuzzy-stranded-NN model strands.

CNN models’ data input setup time (1-minute WAV pre-processing and generation of the input images as well as the Mel spectrogram) lasts about 100 s. From the CNN models, all models except the VGG-16, VGG-19, and WideResNet-40 models have small model memory load times, less than 9 s. VGG-16, VGG-19, and WideResNet-40 increased load times due to the swap usage since their model’s sizes exceed the device available 512 MB RAM (320–415 MB and 2 GB model sizes accordingly). Only the VGG ones can provide device-level inferences from these three models. WideResNet-40 manages to load faster than the VGG models but fails to provide inferences due to memory allocation issues (see Table 5). VGG-16 load times are at least 6.6 times more, and VGG-19 model load times are at least ten times more than the ResNet-50 load times, the best model for provided accuracy results. Therefore, VGG-16 is excluded from further use due to its degraded accuracy results. In contrast, VGG-19 is excluded mainly due to its increased model loading and inference times and not its provided inference accuracies.

Inception_v3 and ResNet-18 load times are 50% and 80% faster than the ResNet-50 model. Nevertheless, their accuracy inference results, especially those achieved by Inception_v3, are disappointing, and it is excluded from further use. The author mentions that the ResNet-18 model with no dropout layers can replace the ResNet-50 model, specifically in IoT devices with limited resources, due to its small model size and fast load and inference executions. Finally, the Half-UNet model presented 45% lower loading times than the ResNet-50 model. However, it showed significant inference times close to 4 times more than those achieved by the ResNet-50 model. Therefore, the Half-UNet model is a slow inference candidate of the ResNet-50 model and, therefore, excluded from edge device computations due to its significant energy consumption to provide an inference result similar to that provided by the ResNet-50 model.

Finally, the detection of CCD conditions provided by the fuzzy-stranded-NN models showed minimal load times (less than 1 s) and mean inference times less than the load times of the ResNet-50 model. Therefore, the fuzzy-stranded-NN models are energy-efficient and require significantly less computational resources to provide inference results. The authors pinpoint that the required resources are single-core 32-bit or 64-bit ARM devices with 32-64MB of available RAM.

To summarize, besides seasonal migrations, beekeeping efforts involve constantly observing the conditions inside the beehive. The main concern of the beekeepers is periodic checks inside the beehives. The time required for such monitoring tasks is significant, and in many cases, it does not lead to any beekeeping actions. For this reason, beekeepers (especially for migratory apiaries) need to use stand-alone IoT periodic condition monitoring systems with built-in intelligent in-hive problem detection capabilities to limit their inspections to only problem-detected hives. To this extent, this paper’s proposition of a new edge computing BeeSD system has been successfully evaluated for detecting swarming and queen loss events in real-time using the ResNet-50 model and catastrophic in-hive environmental conditions using the fuzzy-stranded-NN model. The BeeSD device is capable of autonomous device operation, online and off-line detections, low energy consumption, easy placement in existing Langstroth beehives, and it is equipped with all the necessary technological tools, offering machine and deep learning inferences to detect most critical conditions as a holistic tool to assist the beekeeper. The BeeSD device is also supported by the open-source Thingsboard AS capabilities, a mobile phone web progressive application, and the necessary tools for training models that can then be uploaded over the air to the BeeSD devices.

## 6. Conclusions

This paper presents a new beehive monitoring system supported by a new condition monitoring IoT device called Bee Smart Device, capable of performing edge computing deep learning inferences for detecting swarming events, queen loss, and critical conditions that may lead to Colony Collapse Disorder. The proposed BeeSD device is an improved version previously presented by the author that utilizes multiple temperature probes, a more sensitive and accurate temperature and humidity sensor, and a lavalier microphone. The proposed BeeSD system cloud services and interfaces use the open-source ThingsBoard AS for data visualization and detection notification. Furthermore, specialized dockerized AI containers have been implemented for model training and the ability to perform over-the-air model updates to the BeeSD IoT devices through the ThingsBoard AS.

The BeeSD device uses a CNN deep learning model to detect swarming and queen loss events, taking single-channel images constructed from raw sound recordings as input. Furthermore, specific attributes that indicate growth monitoring, queen tone, thermal stress, and swarming are calculated from specific frequency response values called beegrams, acquired from the Short-Time Fourier Transformations of the BeeSD device audio recordings. For detecting critical CCD conditions, a fuzzy-stranded-NN model takes multi-point temperature and humidity measurements inside the beehive as input. This functionality promotes the BeeSD device as an autonomous holistic detection tool where the most critical events can be detected even in areas of limited network connectivity. The author sets as future work the implementation of a separate communication channel and appropriate mobile phone application user interface and service to provide real-time detections directly to the beekeeper, regardless of the use of the BeeSD gateway, as well as the implementation of a delayed measurement store and forward data logging protocol to the cloud in areas of nonexistent network coverage.

The author experimented with the BeeSD IoT device detection capabilities regarding inference accuracies and execution times. From the experimental results for detecting swarming and queen loss events, the ResNet-50 model presented the best accuracy at 100 training epochs of 99.46%, and fair device-level model load and inference times of 75.32 s. The ResNet-18 model was the fastest among all other tested CNN models, achieving minimum execution inference speeds of 25.2 s, close to the time needed (36 s) for generating beegram measures from multiple 10 s window STFT transformations of recorded audio at the IoT device. Nevertheless, its achieved mean accuracy was close to 93%. The author also points out that the ResNet-18 model can achieve better accuracy if no dropout layers are used. From the author’s experimentation, the mean accuracy results of deep learning models (except the Inception_v3 model) are 1–7% more than the mean accuracy results achieved by machine learning algorithms such as SVM and k-NN for the detection of swarming events and early swarming events. The author also examined even deeper models, such as WideResNet, with even better accuracy results than the ResNet-50 model. Still, significantly more resources were needed regarding size and device memory requirements to provide inference. The author sets further examination of deep CNN models of more than 50Mil parameters executing at the device level, edge computing efforts, and achieved inference accuracies as future work.

The author also experimented with a multi-depth and multi-width neural network model called fuzzy-stranded-NN that utilizes fuzzy beekeeping rules to auto-encode temperature and humidity beehive measurements and offer state detection of conditions favorable to incubating critical CCD events. From the experimental results, it is clear that doubling the number of multi-point temperature measurements inside the beehive can provide 2.5–3% more accurate CCD condition detection results. The author set further evaluation of its multi-probing device by adding other sensory measurements, such as CO_2_ or TVOC, included in his model as future work. Nevertheless, the author pinpoints the beekeeping effort needed to properly place multiple temperature probes inside the beehive. Therefore, the author sets the implementation of a custom beehive with brood box embedded sensors as future work. The author also shows that the fuzzy-stranded-NN strand inference execution times are at least three times faster than the inference times of the ResNet-18 model and can provide accuracy results between 93% and 95%. The author sets as future work the exhaustive experimentation of beehives using multi-point temperature measurements to construct an extensive dataset to enhance further the stranded-NN model’s potential for detecting more accurate CCD events.

## Figures and Tables

**Figure 1 sensors-24-05444-f001:**
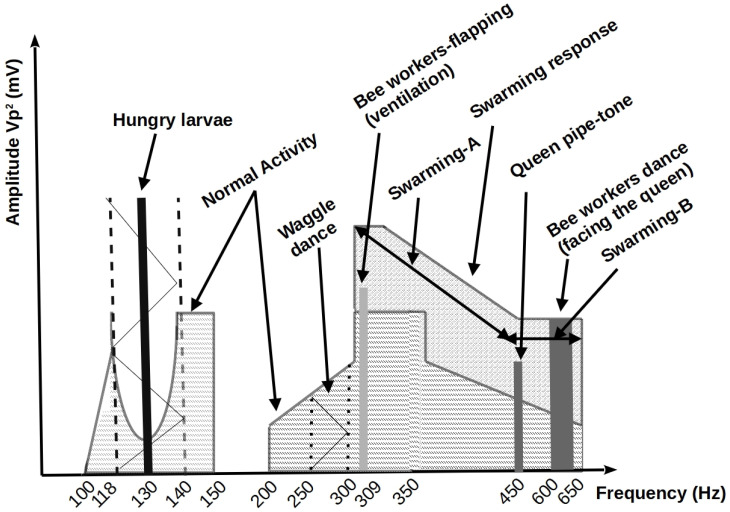
Amplitude response over frequency inside a beehive in normal cases [59,60], of waggle dance [51,60,61], queen pipping tone [62,63], wings flapping under extreme temperature and humidity conditions [62], swarming [9,26,58,64], and due to poor nutrition [62].

**Figure 2 sensors-24-05444-f002:**
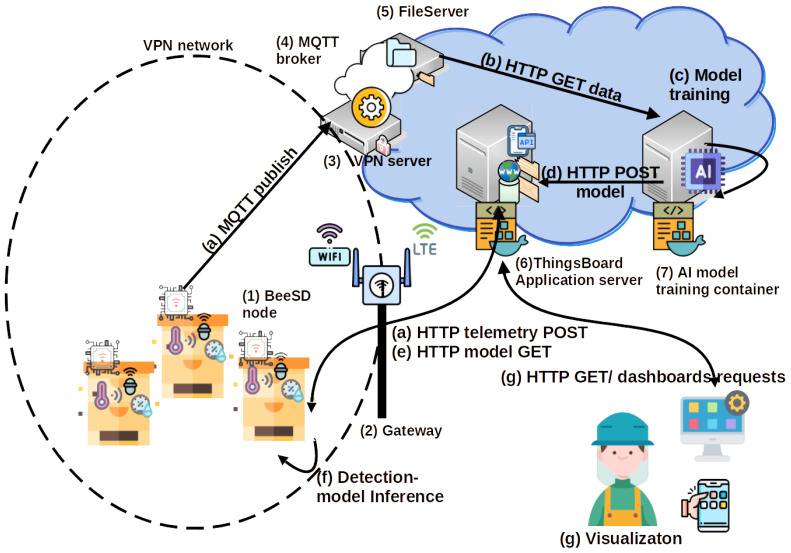
Proposed Bee Smart Detection system’s high-level architecture. Bee Smart Detection system parts, data inputs, outputs, and data processing steps.

**Figure 3 sensors-24-05444-f003:**
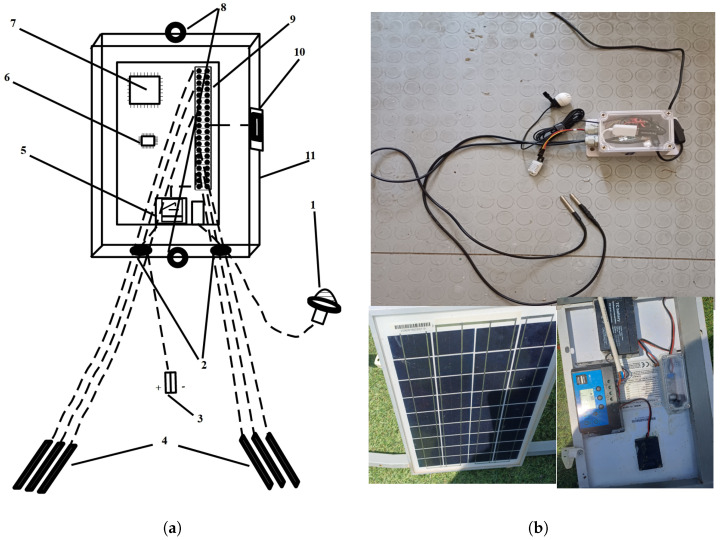
Proposed Bee Smart Detection device, device parts, and connected sensors; (**a**) BeeSD device design; and (**b**) BeeSD device prototype and its external battery source.

**Figure 4 sensors-24-05444-f004:**
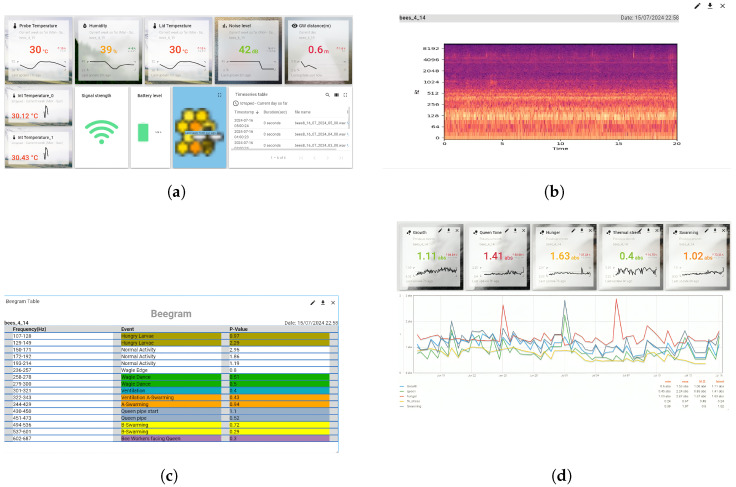
Illustrations of the BeeSD IoT device User Interfaces. These interfaces are presented to the beekeepers via the BeeSD system mobile phone application: (**a**) temperature, humidity, and sound intensity User Interface. (**b**) Mel spectrogram User Interface. (**c**) SFFT contributing frequencies’ mean amplitude values (called beegrams; mentioned in Figure 1) User Interface. (**d**) Aggregated beegram correlation functions values over time corresponding to major indicators (growth, queen tone, hunger, thermal stress, total swarming) User Interfaces.

**Figure 5 sensors-24-05444-f005:**
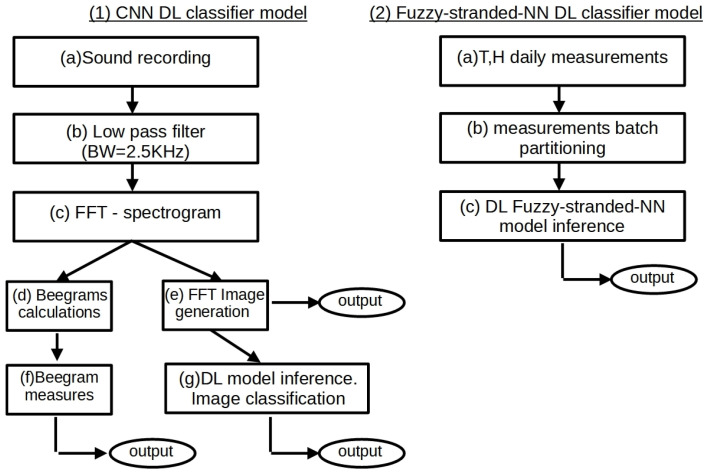
Bee Smart Detection processes for (1) queen loss and swarming (CNN model) and (2) stressful condition events (fuzzy-stranded-NN classifier).

**Figure 6 sensors-24-05444-f006:**
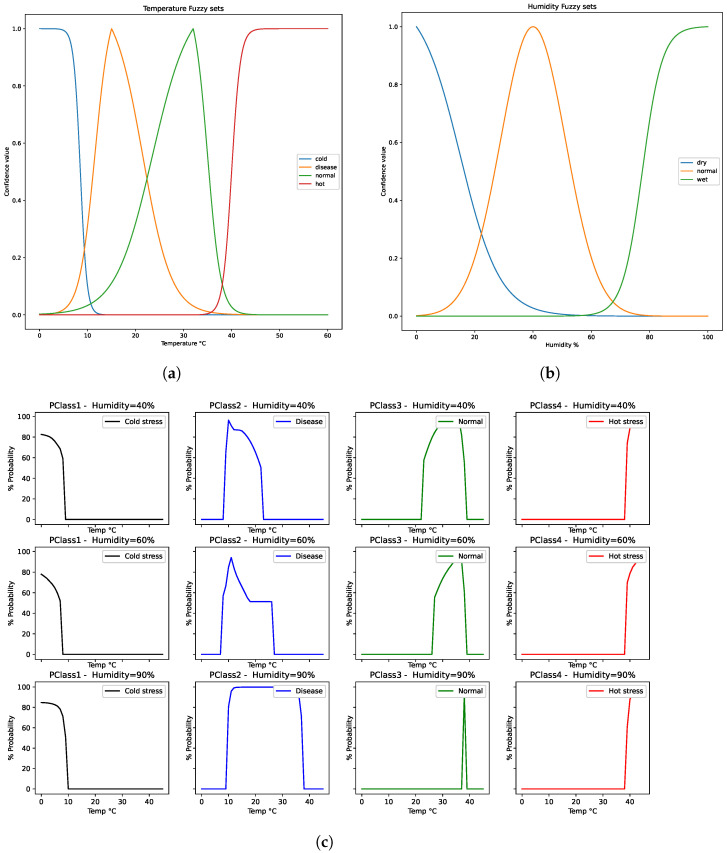
Fuzzification process for data generation for the four fuzzy-stranded-NN classes: (**a**) temperature set, (**b**) humidity set, (**c**) purification process outputs (based on fuzzy rules) in three distinct humidity cases: low (40%), normal (60%), high (90%).

**Figure 7 sensors-24-05444-f007:**
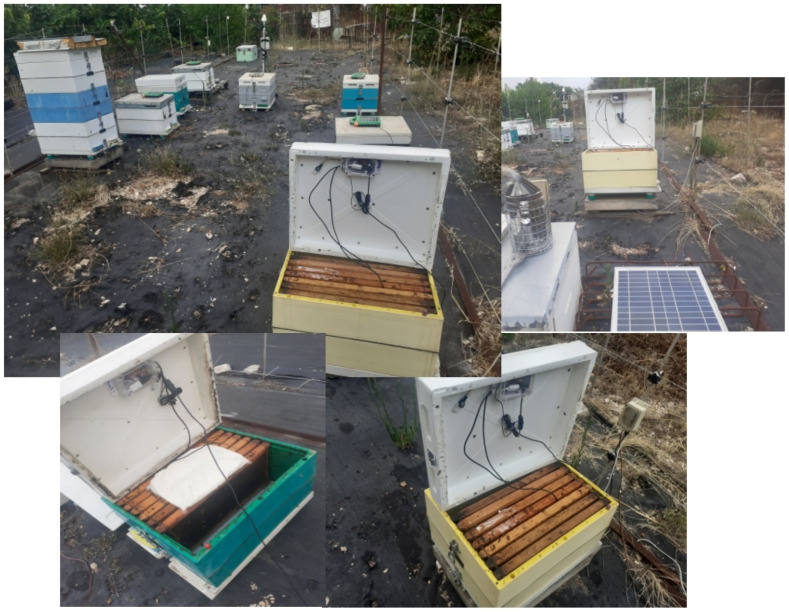
Bee Smart Detection prototypes tested in the beekeeping station of the laboratory team of MicroComputer Systems, Department of Mathematics, University of Ioannina, located in the Ligopsa area, Ioannina, Epirus, Greece.

**Figure 8 sensors-24-05444-f008:**
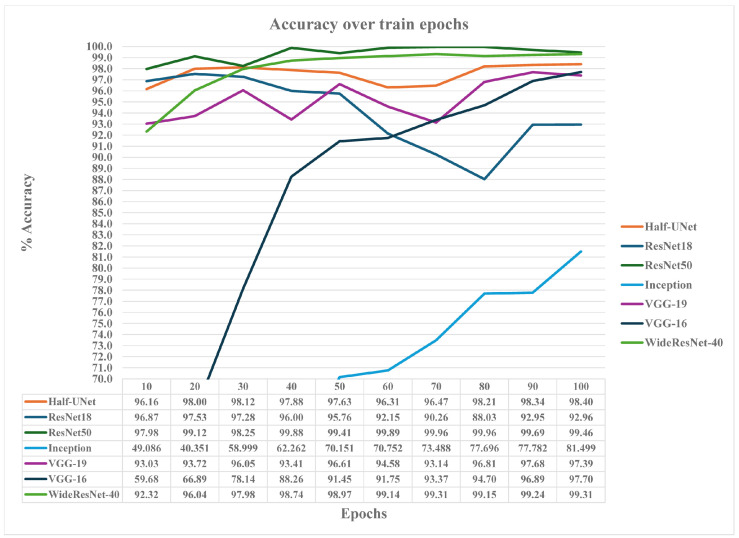
Accuracy over trained epochs for CNN classification models detecting swarming and queen loss.

**Figure 9 sensors-24-05444-f009:**
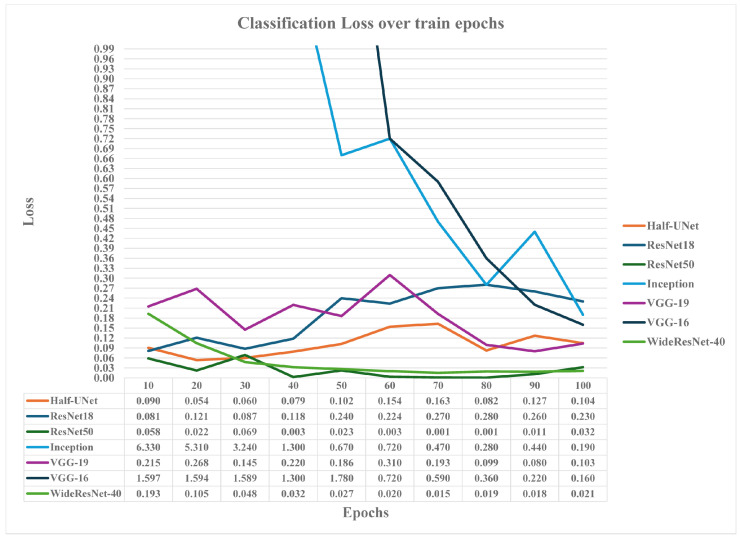
Classification loss over trained epochs for CNN classification models detecting swarming and queen loss.

**Table 1 sensors-24-05444-t001:** Beegram measures calculated from the beegram vector values (attributes).

Beegram Measures	Correlation Functions Calculated from the Beegram Vector $\vec{b}$ Elements	Critical Event Values
Beehive hunger measure (Bh)	$\frac{|\vec{b}\left[0\right]|+|\vec{b}\left[1\right]|}{2}$	Bh≤0.3, Bh>1.8
Beehive growth measure (Bg)	$\frac{1}{6}\sum_{i=2}^{7}\left|\vec{b}\left[i\right]\right|$	Bg≤0.1
Queen tone measure (Qt)	$max\{|\vec{b}\left[11\right]|,|\vec{b}\left[12\right]|\}+|\vec{b}\left[15\right]|$	Qt≤0.3, Qt>0.8
Beehive thermal stress (Th)	$|\vec{b}\left[7\right]|$	$T_{h}>0.5$
Swarming measure (Sw)	$\frac{3}{4}max\{|\vec{b}\left[i\right]{\left|\right\}}_{i=8..10}+$ $\frac{1}{4}max\{|\vec{b}\left[i\right]{\left|\right\}}_{i=11..14}$	$S_{w}>0.7$

**Table 2 sensors-24-05444-t002:** Fuzzy inference rules of the fuzzy-stranded-NN model.

Fuzzy Sets	Humidity Dry	Humidity Normal	Humidity Wet
Temperature cold	minus (class-0)	class-0	plus (class-0)
Temperature disease	class-2	class-1	plus (class-1)
Temperature normal	class-2	plus (class-2)	minus (class-1)
Temperature hot	plus (class-3)	class-3	minus (class-2)

**Table 3 sensors-24-05444-t003:** Fuzzy-stranded-NN strands of 96, 144 batch sizes and CNN model sizes, number of trainable parameters, and cloud inference speeds sorted by total loadable model sizes.

Model	Input Size	No. Trainable Params (Mil)	Params Size (MB)	Total Size (MB)	Cloud Inference Speed (s)
Stranded-NN-96 (1d-1h-4mp)	(96,)	0.013	0.054	<1 MB	0.015
Stranded-NN-144 (1d-1h-6mp)	(144,)	0.046	0.221	<1 MB	0.019
ResNet-18	(224, 224)	11.691	44.60	106.25	0.024
Inception_v3	(224, 224)	19.534	74.52	150.72	0.057
ResNet-50	(224, 224)	21.280	81.18	177.65	0.040
Half-UNet	(224, 224)	19.636	74.91	441.86	0.084
VGG-16	(224, 224)	134.288	512.27	834.53	0.117
VGG-19	(224, 224)	139.600	532.53	884.53	0.139
WideResNet-40	(224, 224)	34.13	35.12	1939.70	0.338

**Table 4 sensors-24-05444-t004:** CNN and fuzzy-stranded-NN model accuracy and convergence speed metric values over training epochs.

Model	Accuracy|Conv. 20 Epochs	Accuracy|Conv. 50 Epochs	Accuracy|Conv. 100 Epochs
VGG-16	58.52|1.22	72.02|0.73	83.04|0.12
VGG-19	88.90|2.14	92.32|0.02	92.75|0.011
Inception_v3	53.62|1.48	57.52|0.23	63.53|0.295
Half-UNet	96.06|0.33	96.84|−0.18	96.31|0.14
ResNet-18	95.89|0.47	95.71|−0.034	93.88|−0.11
ResNet-50	97.79|0.16	98.79|0.0016	99.26|0.006
WideResNet-40	96.03|0.52	97.52|0.08	98.41|0.044
Stranded-NN-96 (1d-1h-4mp)	78.5|1.56	88.7|0.92	93.6|0.02
Stranded-NN-144 (1d-1h-6mp)	84.6|1.78	94.6|0.45	95.2|0.01

**Table 5 sensors-24-05444-t005:** Edge computing CNN models and fuzzy-stranded-NN load and inference times using a quad-core 64-bit ARM-CPU.

Model	Data Transform Time (s)	Model Load Time (s)	Inference Time (s)
Inception_v3	99.932	4.6	50.214
Half-UNet	99.932	5.6	234.778
ResNet-18	99.932	1.59	23.616
ResNet-50	99.932	8.2	67.153
WideResNet-40	99.932	48.2	allocation issues at least 1.2 GB of RAM required
VGG-16	99.932	63.1	285.7
VGG-19	99.932	97.5	421.5
Stranded-NN-96	-	0.8	6.2
Stranded-NN-144	-	0.8	8.6

## Data Availability

No new data were created.

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
