# Peer review of "Beehive Smart Detector Device for the Detection of Critical Conditions That Utilize Edge Device Computations and Deep Learning Inferences"

_sensors, 2024, doi:10.3390/s24165444_

Round 1

Reviewer 1 Report

Comments and Suggestions for Authors

The goal of the paper is to present a new IoT-based edge device called the Bee Smart Detector (BeeSD), designed for monitoring beehives and detecting critical conditions such as swarming, queen loss, and environmental stressors that could lead to Colony Collapse Disorder (CCD). The device utilizes advanced edge computing and deep learning algorithms, specifically Convolutional Neural Networks (CNNs) and fuzzy multi-layered neural networks, to analyze sensory data from temperature, humidity, and sound measurements within the hive. The system aims to provide accurate, real-time monitoring and early detection of catastrophic events to support beekeeping practices and enhance hive management.

The paper highlights the innovative integration of IoT and deep learning technologies in beekeeping, providing a practical solution to enhance hive health and management.

Below are some comments in order to improve your paper: 

  • Provide more background on the importance of hive health and the impact of CCD. Mention previous approaches and their limitations.
  • Clearly state the main objective of your study. For example: "The objective of this study is to develop and evaluate a real-time edge-device for monitoring critical conditions in beehives using advanced deep learning techniques."
  • Compare your work with the latest research more explicitly. Highlight the advancements your system brings in comparison to existing technologies.
  • Provide detailed technical specifications of the BeeSD device components. Use diagrams to illustrate the setup.
  • Explain the rationale for choosing specific deep learning models like ResNet-50 and fuzzy-stranded-NN. Provide a brief description of each model and its relevance to the task.
  • Elaborate on the dataset collection process, including the conditions under which data was collected, the volume of data, and any preprocessing steps.
  • Detail the training process, including any data augmentation techniques used, the number of epochs, batch size, and computational resources. Discuss the validation process and metrics used to evaluate the models.
  • Provide a rationale for selecting specific metrics. Explain how these metrics are calculated and why they are appropriate for the study.
  • Discuss the practical implications of your findings for beekeepers. Explain how the device can be implemented in real-world scenarios and its potential impact on hive health management.
Comments on the Quality of English Language

The English in the paper is generally clear and well-structured, but there are a few areas that could be improved for better clarity and readability.

Author Response

Author letter of amendments.

The goal of the paper is to present a new IoT-based edge device called the Bee Smart Detector (BeeSD), designed for monitoring beehives and detecting critical conditions such as swarming, queen loss, and environmental stressors that could lead to Colony Collapse Disorder (CCD). The device utilizes advanced edge computing and deep learning algorithms, specifically Convolutional Neural Networks (CNNs) and fuzzy multi-layered neural networks, to analyze sensory data from temperature, humidity, and sound measurements within the hive. The system aims to provide accurate, real-time monitoring and early detection of catastrophic events to support beekeeping practices and enhance hive management.

The paper highlights the innovative integration of IoT and deep learning technologies in beekeeping, providing a practical solution to enhance hive health and management.

Below are some comments in order to improve your paper: 

Response: Thank you for your time and effort in reviewing our manuscript. Here, we quote our responses and amendments performed based on your comments.  

Comment 1: Provide more background on the importance of hive health and the impact of CCD. Mention previous approaches and their limitations.

Response: Section 2 first paragraph has been revised (CCD causes). An additional paragraph with references has been added to section 2 (paragraph 2), explicitly mentioning the main causes of CCD events.  

Comment 2: Clearly state the main objective of your study. For example: "The objective of this study is to develop and evaluate a real-time edge device for monitoring critical conditions in beehives using advanced deep learning techniques."

Response: Thank you very much for your remark. The last two paragraphs of the Introduction have been amended to present this paper's objective.

Comment 3: Compare your work with the latest research more explicitly. Highlight the advancements your system brings in comparison to existing technologies.

Response: The paragraph in line 865 has been rewritten to highlight the BeeSD device's advancements. Two paragraphs have been added after this paragraph.

Comment 4: Provide detailed technical specifications of the BeeSD device components. Use diagrams to illustrate the setup.

Response: Figure 3. a and subsection 3.2, paragraphs 1,2, present the BeeSD device parts in detail. The cloud components of the BeeSD device are described in subsection 3.1. Appropriate amendments have been made to also refer to Figure 4 (mobile phone visualization dashboard illustrations)

Comment 5: Explain the rationale for choosing specific deep learning models like ResNet-50 and fuzzy-stranded-NN. Provide a brief description of each model and its relevance to the task.

Response: The experimental scenario section has been expanded by adding an additional first paragraph. Paragraphs 2 and 4 have also been added to provide a rationale for choosing.

Comment 6: Elaborate on the dataset collection process, including the conditions under which data was collected, the volume of data, and any preprocessing steps.

Response: Section 5 (first paragraph) has been added to explain how the dataset was collected, the data volume, and the preprocessing steps (prior to training).

 Comment 7: Detail the training process, including any data augmentation techniques used, the number of epochs, batch size, and computational resources. Discuss the validation process and metrics used to evaluate the models.

Response: Experimental scenario paragraphs 3 (amended) and 5 (added) to provide training hyperparameters used. The metrics used are presented at 4.1

Comment 8: Provide a rationale for selecting specific metrics. Explain how these metrics are calculated and why they are appropriate for the study.

Response: An appropriate paragraph has been added at the beginning of section 4.1 to provide a rationale. A description of the models' calculation metrics used is included in 4.1   

Comment 9: Discuss the practical implications of your findings for beekeepers. Explain how the device can be implemented in real-world scenarios and its potential impact on hive health management.

Response: Two paragraphs have been added at the end of the section (last two paragraphs), and an appropriate figure has been added in section 5, (Experimental scenario) showing the BeeSD prototypes in action (placed in beehives).

Comments on the Quality of English Language

The English in the paper is generally clear and well-structured, but there are a few areas that could be improved for better clarity and readability.

Response: The author made typo errors and syntactical and grammatical corrections to improve his manuscript's readability.

Reviewer 2 Report

Comments and Suggestions for Authors

This article introduces the construction of an IoT monitoring system for the beekeeping industry, as well as a detection method based on deep learning.

the paper's description is overly verbose, with a lack of concise language. In the deep learning section, experiments using different deep learning methods for the same problem are compared. However, a comparison with the results of similar work by peers is absent.

Overall, the paper exhibits strong engineering practicality, providing detailed descriptions of the system's construction and the parameters of each module, which serves as a valuable reference for information technology in the beekeeping industry. Nevertheless, there is room for improvement in refining and summarizing the research questions and methodologies.

Recommendations:

1. Revise the abstract. The current abstract is too long, with vague mentions of innovation points and methods.

2. Include a comparison with similar work by peers to provide a broader context for the research.

3. Refine the language throughout the paper to make it more concise and focused.

Author Response

Author letter of amendments

This article introduces the construction of an IoT monitoring system for the beekeeping industry, as well as a detection method based on deep learning.

the paper's description is overly verbose, with a lack of concise language. In the deep learning section, experiments using different deep learning methods for the same problem are compared. However, a comparison with the results of similar work by peers is absent. Overall, the paper exhibits strong engineering practicality, providing detailed descriptions of the system's construction and the parameters of each module, which serves as a valuable reference for information technology in the beekeeping industry. Nevertheless, there is room for improvement in refining and summarizing the research questions and methodologies.

Response: Thank you for your time and effort in reviewing our manuscript. Here, we quote our responses and amendments performed based on your comments.  

Recommendations:

Comment 1: Revise the abstract. The current abstract is too long, with vague mentions of innovation points and methods.

Response: The abstract has been revised so as to be more concise and focused.

Comment 2: Include a comparison with similar work by peers to provide a broader context for the research.

Response: Two additional paragraphs have been added after the paragraph at line 865 that include a comparison with similar work

Comment 3: Refine the language throughout the paper to make it more concise and focused.

Response: The author made appropriate refinements and syntactical and grammatical corrections to improve his manuscript's readability

Reviewer 3 Report

Comments and Suggestions for Authors

In this work the authors propose a BeeSD system, which is a cutting-edge solution for monitoring beehives using edge computing and deep learning. It employs IoT devices with temperature and humidity sensors, along with a lavalier microphone, to collect data inside the beehive. The system operates in two modes: training, where it gathers raw data for model development, and detection, where it analyzes real-time sensory inputs to identify critical events like swarming, queen loss, and starvation. Data is processed on-site using CNN and fuzzy-stranded-NN models, and results are transmitted to a ThingsBoard server for visualization and analysis. A dedicated gateway ensures continuous operation, while a mobile app provides beekeepers with real-time updates and alerts.

The paper suggests that dropout layers in ResNet-18 models might be counterproductive, but it lacks a thorough investigation into why dropout affects performance or how alternative regularization techniques could be employed. It is recommended to clearly mention why dropout layers are better than regularization techniques for their application use case.

There should be a discussion on the impact of data quality on the model's performance, since the data is collected in a very dynamic environment. This environment is highly prone to surrounding noise, which can lead to variability on model performance. This variability/uncertainity in environmental conditions and sound recording quality can affect the robustness of the models, which should be explained.

It is also recommended to explore other existing beekeeping solutions in related works that utilize IoT technologies for e.g.:
https://ieeexplore.ieee.org/document/9040593

https://www.sciencedirect.com/science/article/pii/S2772375523001600

Author Response

Author letter of amendments

In this work the authors propose a BeeSD system, which is a cutting-edge solution for monitoring beehives using edge computing and deep learning. It employs IoT devices with temperature and humidity sensors, along with a lavalier microphone, to collect data inside the beehive. The system operates in two modes: training, where it gathers raw data for model development, and detection, where it analyzes real-time sensory inputs to identify critical events like swarming, queen loss, and starvation. Data is processed on-site using CNN and fuzzy-stranded-NN models, and results are transmitted to a ThingsBoard server for visualization and analysis. A dedicated gateway ensures continuous operation, while a mobile app provides beekeepers with real-time updates and alerts.

Response: Thank you for your time and effort in reviewing our manuscript. Here, we quote our responses and amendments performed based on your comments.  

Comment 1: The paper suggests that dropout layers in ResNet-18 models might be counterproductive, but it lacks a thorough investigation into why dropout affects performance or how alternative regularization techniques could be employed. It is recommended to clearly mention why dropout layers are better than regularization techniques for their application use case.

Response: The paragraph at line 841 has been revised to clearly mention why dropout layers at the ResNet-18 model have been used.

Comment 2: There should be a discussion on the impact of data quality on the model's performance since the data is collected in a very dynamic environment. This environment is highly prone to surrounding noise, which can lead to variability in model performance. This variability/uncertainty in environmental conditions and sound recording quality can affect the robustness of the models, which should be explained.

Response: The data collection environment (beekeeping station information location and station image) has been added to the first paragraph of the experimental scenario section. All observations (data) have been collected from this station. The station's location has been selected to be in a secluded area away from environmental noises that may affect the experimentation. Additionally, all sound recordings have been performed at night to avoid animal interference. You may see our beekeeping station from Google Maps (beekeeping station location information is also provided in this paragraph).
Comment 3: It is also recommended to explore other existing beekeeping solutions in related works that utilize IoT technologies for e.g.:
https://ieeexplore.ieee.org/document/9040593

https://www.sciencedirect.com/science/article/pii/S2772375523001600

Response: Additional references have been added in lines 263-293 of the related work section, mentioning other existing beekeeping solutions

Round 2

Reviewer 1 Report

Comments and Suggestions for Authors

The authors have made significant improvements to the paper and has addressed all of my concerns.

Comments on the Quality of English Language

The author has made significant improvements to the paper 

Author Response

Thank you for your comments, time, and effort in reviewing my manuscript.

Best regards

Sotirios Kontogiannis 

Reviewer 2 Report

Comments and Suggestions for Authors

The paper has made revisions to all the proposed suggestions. 

This paper has strong engineering practicality and provides valuable insights for the beekeeping industry with IoT.

The conclusion should be further refined, avoiding the listing of experimental data and results.And it is better to discribe future research work.

Author Response

Thank you for your comments, time, and effort in reviewing my manuscript.

The conclusions section has been refined as instructed

Best regards

Sotirios Kontogiannis 

Reviewer 3 Report

Comments and Suggestions for Authors

 the comments have been addressed in the revised manuscript.

Author Response

(The authors gave the same response as above.)
